. Pathogens

# Tibetan PHD2$^{D4E;C127S}$ variant protects from viral diseases in hypoxia, but predispose to infections in normoxia via HIFα:IFN axis

Riya Ghosh[1,2], Garima Joshi[2], Nishith M. Shrimali[2], Kanchan Bhardwaj[1,2], Tsewang Chorol[3], Tashi Thinlas[3], Parvaiz A. Koul[4], Josef T. Prchal[5], Prasenjit Guchhait[2]*

1 Manav Rachna International Institute of Research and Studies, Faridabad, India, 2 Regional Centre for Biotechnology, National Capital Region Biotech Science Cluster, Faridabad, India, 3 Sonam Nurboo Memorial Hospital, Leh-Ladakh, Jammu and Kashmir, India, 4 Department of Internal and Pulmonary Medicine, Sher-i-Kashmir Institute of Medical Sciences, Srinagar, India, 5 Department of Medicine, University of Utah School of Medicine & Huntsman Cancer Center and George E. Wahlin Veteran's Administration Medical Center, Salt Lake City, Utah, United States of America

* prasenjit@rcb.res.in

## Abstract

We previously reported that Tibetan-specific variant of prolyl-hydroxylase-2 (PHD2)$^{D4E;C127S}$ protects highlanders from hypoxia-triggered pathologies by destabilizing hypoxia-inducible factor (HIF)-1α. Others have reported that stabilized HIF1α negatively regulates interferon (IFN)-regulating factor (IRF)-3 under hypoxia. We examined the role of PHD2$^{D4E;C127S}$ variant in IFN synthesis in immune cells during viral infections. Primary monocytes and cells engineered to express the PHD2$^{D4E;C127S}$ variant displayed protection against dengue virus (DENV)-2 infection by suppressing HIF1α, in turn promoting IRF-3 and IFNα/β synthesis in hypoxia (3% $O_2$) in vitro. However, under normoxia (21% $O_2$), these mutant cells increased reactive oxygen species (ROS) generation following DENV2 infection. Increased ROS then suppressed PHD2$^{D4E;C127S}$ activity, reflected by reduced hydroxylation and concomitant stabilization of HIF1α, resulting in suppressed IFN synthesis and higher DENV2 replication. The PHD2$^{WT}$ cells demonstrated the opposite trend. Our data further confirmed the inverse correlation between HIF1α and IFN pathways. CAY10585, a HIF1α-inhibitor, decreased the DENV2 infection by increasing *IFN-A/B* and *IRF-3/7/9* expression. HIF1α-depleted monocytes also showed a similar response to the infection. We extended our findings to COVID-19 infection. The CD4/CD8 T-cells collected from Tibetans with PHD2$^{D4E;C127S}$ variant and exposed to SARS-CoV-2 infection showed elevated expression of IFN-γ in response to exposure to SARS-CoV-2 receptor-binding domain (RBD) peptide under hypoxia, and a lesser expression under normoxia. The study thus highlights a unique crosstalk of PHD2$^{D4E;C127S}$ variant with HIF1α-IFN axis under environmental $pO_2$ in protecting or predisposing Tibetans to viral infections.

**Data availability statement:** All data are mentioned in the manuscript. Raw data for all Figures are attached as Zip file.

**Funding:** BT/PR22881 and BT/PR22985 from the Department of Biotechnology, Govt. of India to P.G. The funders had no role in study design, data collection and analysis, decision to publish, or preparation of the manuscript.

**Competing interests:** The authors have declared that no competing interests exist.

## Author summary

Approximately 400 million people permanently reside in high mountain ranges. One such highlander population, Tibetans, who developed genetic adaptation against hypobaric hypoxia and mountain maladies. Earlier we have described that the Tibetan carrying PHD2$^{D4E;C127S}$ variant are protected from hypoxia induced clinical symptoms like polycythemia and edema. The PHD2$^{D4E;C127S}$ variant has significantly higher affinity for oxygen, in turn degrades HIF1α/ HIF2α more efficiently than the PHD2$^{WT}$. We describe in this study for the first time that the immune response of PHD2$^{D4E;C127S}$ or PHD2$^{WT}$ monocytes/T cells against viral infections under hypoxia vs normoxia. Study describes a unique crosstalk of PHD2$^{D4E;C127S}$ variant with HIF1α under environmental pO$_2$ in regulating the anti-viral interferon (IFN) response against the viral infections including dengue and COVID-19. Tibetan PHD2$^{D4E;C127S}$ variant protects both viral diseases in hypoxia, but predisposes to infections in normoxia.

## Introduction

Approximately 400 million people permanently reside in high mountain ranges. Further, millions of trekkers and climbers travel to high-altitudes, miners and soldiers are deployed for duty in these remote areas. High-altitude environments display distinctive challenges to individuals. At high-altitudes above 2500 meters, the body is exposed to a lower barometric pressure, resulting in a decreased partial pressure of oxygen (pO$_2$), increased UV radiation, less humidity, and a higher exposure to extreme weather. These harsh environmental conditions interact in a complex way to alter the host immune response, leading to a higher susceptibility to infection. Respiratory diseases, such as pneumonia, are common at high-altitude among the travellers. It has also been suggested that pulmonary infection can make an individual more susceptible to acute mountain sickness [1]. High-altitude has been found to be a risk factor associated with Respiratory syncytial virus (RSV) infection in travellers [2]. Hepatitis A, B, C, and E virus infections are common in travellers at high-altitudes [3]. Dengue and Chikungunya viruses are endemic to the tropics and sub-tropic Himalayan altitudes [4]. Typhoid fever, also known as enteric fever, is one of the most common causes of undifferentiated febrile illness in the Indian subcontinent. Carrion's disease is a febrile condition endemic to arid, high-altitude (approximately 600–3,200 m) valleys in the Andes Mountains of Peru, Colombia, and Ecuador.

Under hypoxia, stabilization of hypoxia-inducible factors (HIFs) regulates several immune response genes. HIF-1 and HIF-2 (dimers of α and β subunit) play essential role in the cellular response to low oxygen, orchestrating a metabolic switch that allows the cells, including immune cells, to survive in this environment. The innate immune cells including neutrophils, macrophages, mast cells, dendritic cells, and natural killer cells are known for their unique responses under hypoxia. The adaptive or acquired immune response, mediated by T and B lymphocytes are known to be affected by the hypoxia [5,6].

The epidemiological data reveal that high altitude natives in countries like Nepal and Bhutan were less susceptible to the COVID-19 infection compared to those living in lower altitudes [7]. The better ability of high-altitude natives to protect against the attack of this deadly virus could be due to several genetic and physiological adaptations or it could also be attributed to the environmental conditions existing at high-altitude [8].

As of yet no study clearly defined the mechanism of immune response against viral infections in populations such as Tibetans, Ethiopians and Andeans, who developed evolutionary genetic adaptation to environmental hypoxia [9–16]. In Tibetans, genetic adaptation to high altitude has been primarily associated with the genes *EPAS1* [encodes hypoxia inducible factor (HIF)2α] and *EGLN1* [encodes prolyl hydroxylase 2 (PHD2), a negative regulator of HIFs], have undergone the strongest evolutionary genetic selection [10,12,13,17]. PHD2$^{D4E;C127S}$ variant alone, or in combination with HIF2α variant, is associated with some degree of protection against elevated haemoglobin concentration in Tibetans [9,11,13,15]. We reported that missense mutations c.[12C>G; 380G>C] in *EGLN1* (PHD2$^{D4E;C127S}$) have about ~85% gene frequency in Tibetans and protects them from polycythaemia in hypoxic environment of high-altitude [11]. We have shown that the protein of PHD2$^{D4E;C127S}$ variant has significantly higher affinity for oxygen than the wild type (PHD2$^{WT}$) and thus degrades HIF1α and HIF2α more efficiently. Thus, inhibition of HIF stabilization in a hypoxic environment due to the enhanced activity of PHD2$^{D4E;C127S}$ variant is one of the primary facilitators of Tibetan adaptation to high altitude [11].

In this study, we describe a unique crosstalk of PHD2$^{D4E;C127S}$ variant with environmental pO2 in protecting or predisposing to viral infections by modulating the HIF1α-interferon (IFN) axis.

## Results

### Monocytes from PHD2$^{D4E;C127S}$ Tibetans have higher DENV2 infection in normoxia, but lower in hypoxia as compared to PHD2$^{WT}$ in vitro

To investigate the role of the Tibetan specific PHD2$^{D4E;C127S}$ variant in viral infections, we pre-exposed the primary monocytes, isolated from the PBMCs of PHD2$^{D4E;C127S}$ or PHD2$^{WT}$ individuals to hypoxia (3% $O_2$) or normoxia (21% $O_2$) for 8 hrs, and infected with dengue virus (DENV)-2 with a MOI~3 and continued to incubate the cells under same pO2 environment for 16 hrs. The PHD2$^{D4E;C127S}$ monocytes showed more viral RNA copies (Fig 1A) and dengue NS1 (Fig 1B and 1C) expression in normoxia as compared to PHD2$^{WT}$ monocytes. Conversely, PHD2$^{D4E;C127S}$ monocytes had lesser viral RNA copies (Fig 1A) and NS1 (Fig 1B and 1C) expression in hypoxia than PHD2$^{WT}$.

### The primary PHD2$^{D4E;C127S}$ monocytes have lower IFNα/β synthesis in normoxia, but higher in hypoxia as compared to PHD2$^{WT}$ in vitro

PHD2$^{D4E;C127S}$ monocytes infected with DENV2 have lower expression of *IFNA1* and *IFNB1* (Fig 1D and 1E), and released less IFNα and IFNβ (Fig 1F and 1G) in normoxia. A higher expression of all the immune markers were observed in hypoxia (Fig 1D–1G) as compared to PHD2$^{WT}$. The expression of interferon regulatory factors *IRF3*, *IRF7* and *IRF9* genes also have similar trends (Fig 1H–1J).

### HIF1α expression inversely correlated with IFNα/β synthesis in these monocytes

HIF1α, primary transcription factor regulating the hypoxia responses, has decreased expression in monocytes expressing PHD2$^{D4E;C127S}$ variant under hypoxia as compared to PHD2$^{WT}$ (Fig 1K and 1L, flow cytometry data; Fig 1M and 1N, fluorescence microscopy data), suggesting an inverse correlation with IFNα/β axis.

### U937 monocytic cells engineered with *EGLN1*$^{c.[12C>G;380G>C]}$ or *EGLN1*$^{WT}$ had similar responses as the primary cells, underlying the inverse relationship between HIF1α and IFNα/β axis in DENV2 infection

To investigate details of the above observations, we used engineered PHD2$^{D4E;C127S}$(*EGLN1*$^{c.[12C>G; 380G>C]}$) and PHD2$^{WT}$(*EGLN1*$^{WT}$) U937 monocytic cell line. Generation of these cells is described in Fig 2A and 2B. Elevated DENV2

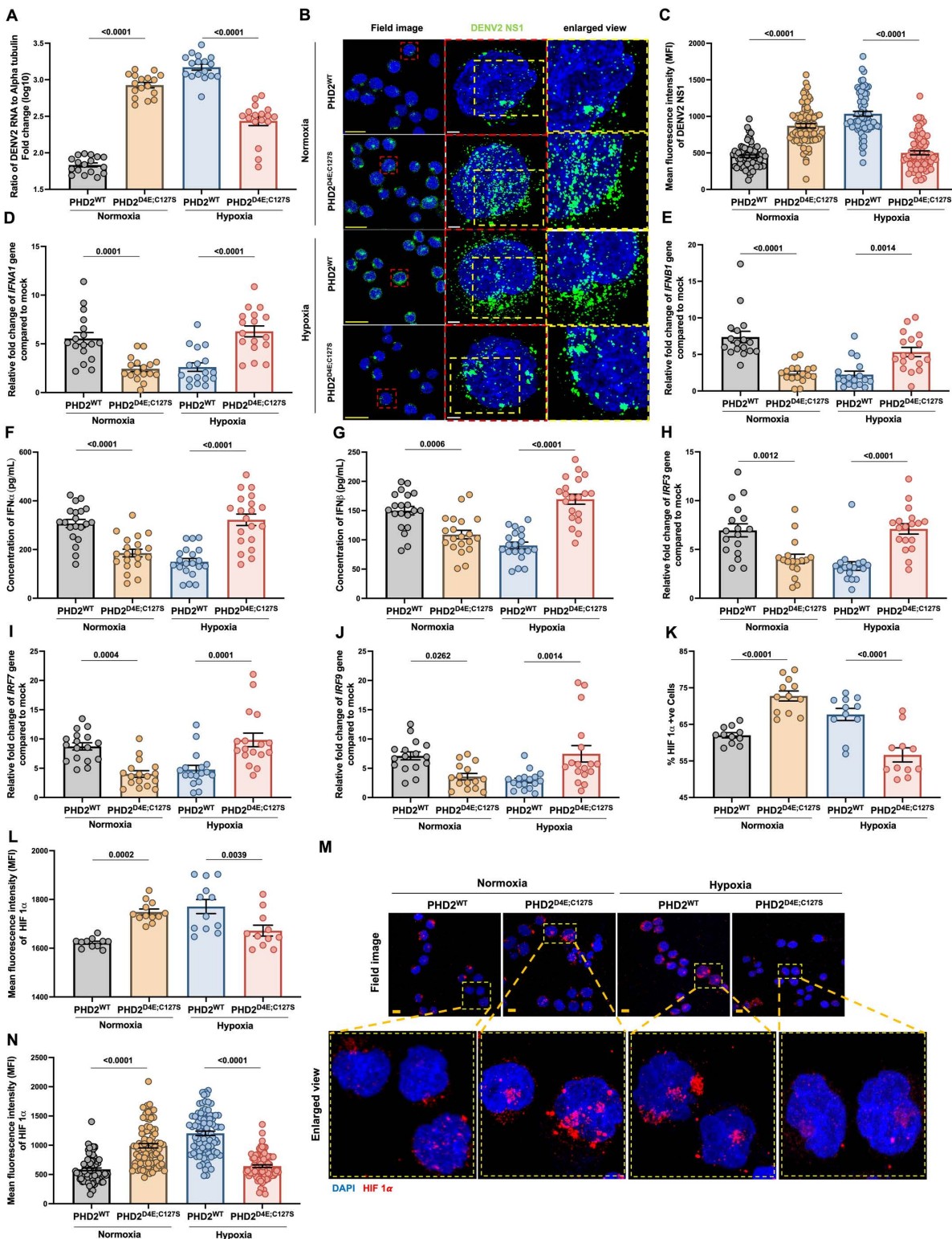

**Fig 1. DENV2 infection in primary monocytes expressing PHD2$^{D4E;C127S}$ or PHD2$^{WT}$ of Tibetan individuals in vitro under normoxia or hypoxia.**
Primary monocytes from Tibetans with either PHD2$^{WT}$ or PHD2$^{D4E;C127S}$ were pre-exposed under normoxia (21% $O_2$) or hypoxia (3% $O_2$) for 8 hrs. Followed by cells were infected with DENV2 (MOI~3) and incubated for another 16 hrs. **A.** Cell pellets were used for measurement of DENV2 RNA levels

using qRT-PCR. Presented as relative fold-change (log10) of the ratio of DENV2 RNA to human α-Tubulin. One-way ANOVA and Bonferroni's post-test were used for analysis. Dot represents one individual value. **B.** Cells from the above experiment were fixed and stained for DENV2 NS1 (green) and nucleus (DAPI, blue). **C.** Data are the mean fluorescence intensity (MFI) of NS1 and was calculated as mentioned above. **D.** Similarly, IFN*A1* and **E.** IFN*B1* levels in cell pellets were measured using qRT-PCR, relative fold-change after normalization with human α-Tubulin, as compared to mock. One-way ANOVA and Bonferroni's post-test were used for analysis. Single dot represents one individual value. **F.** IFNα and **G.** IFNβ concentrations in cell supernatant were measured using ELISA. **H.** Relative fold change of IRF3, **I.** IRF7 and **J.** IRF9 genes were measured by qRT-PCR from cell pellets and calculated as mentioned above. K-L. The HIF1α positive monocytes were measured from above experiment using flow cytometry. Data are calculated as mentioned above and presented as percentage as well as MFI. Flow cytometry gating strategy is mentioned in S1A Fig. **M.** Cells from the above experiment were fixed and stained for HIF1α (red) and nucleus (DAPI, blue). **N.** MFI was calculated. Data are mean ± SEM and P-values are mentioned in graphs.

RNA level (Fig 2C) and dengue NS1 (Fig 2D, 2E and 2M) were observed in *EGLN1*[c.[12C>G; 380G>C]] cells in normoxia (21% $O_2$), but infection was decreased in hypoxia (3% $O_2$) as compared to *EGLN1*[WT] cells. Inversely, the suppressed *IFNA1* and *IFNB1* as well as IFNα and IFNβ (Fig 2F–2I) along with *IRF3*, *IRF7* and *IRF9* genes (Fig 2J–2L) were observed in *EGLN1*[c.[12C>G; 380G>C]] cells as compared to *EGLN1*[WT] in normoxia. The opposite trends were observed in hypoxia. However, other inflammatory cytokines such as TNFα, IL6 and IL1ß did not alter significantly between these phenotypes upon DENV2 infection under normoxia or hypoxia (S4A–S4F Fig).

Immunoblot data showed the increased HIF1α and HIF2α expression, in parallel with higher DENV NS1 expression in *EGLN1*[c.[12C>G; 380G>C]] cells in normoxia, but lower in hypoxia as compared to *EGLN1*[WT] (Fig 2M, densitometry data are mentioned in S2A–S2C Fig).

### Elevated ROS correlated with decreased PHD2 activity and increased HIF1α stabilization

To investigate the mechanism of HIF1α elevation in the *EGLN1*[c.[12C>G; 380G>C]] cells upon viral infection under normoxia, we found an elevated reactive oxygen species (ROS), both total (Fig 3A) and also mitochondrial (Fig 3B). Elevated ROS might have inhibited the PHD2[D4E;C127S] activity to stabilise HIF1α as described [18,19]. Further, we measured the hydroxylated HIF1α and total HIF1α. Interestingly, the elevated total HIF1α showed parallel correlation with ROS levels in mutant cells at 24 hrs after viral infection under normoxia (Fig 3A–3E). While, the hydroxylated HIF1α showed inverse correlation with the ROS in these mutant cells after 12 and 18 hrs infection under normoxia, indicating an inhibitory effect of ROS on the PHD2 that hydroxylate HIF1α (Fig 3F–3I).

### The HIF1α inhibitor improved the IFNα/β synthesis and inhibited DENV2 infection in vitro

To evaluate the role of HIF1α in interferon synthesis, we used HIF1α inhibitor CAY10585, which significantly reduced viral RNA levels in both *EGLN1*[c.[12C>G;380G>C]] and *EGLN1*[WT] cells in normoxia (Fig 4A). CAY10585 increased phosphorylated (P)-IRF3 (Fig 4B, densitometry analysis is depicted in S2D–S2F Fig). CAY10585 increased *IFNA1*, *IFNB1* gene expression (Fig 4C and 4D), and secretion of IFNα and IFNβ (Fig 4E and 4F) along with *IRF3*, *IRF7* and *IRF9* genes (Fig 4G–4I). We validated the inhibitory effects of CAY10585 on HIF1α, showing the abrogation of expression of a HIF1α-specific target gens like VEGF (S3A–S3B Fig).

### The HIF1α-deficient cells have lower DENV2 infection and elevated IFNα/β expression

To confirm the above observations, we used HIF1α-KD (*HIF1A*-shRNA mediated depletion) U937 cells. Knockdown of HIF1α was confirmed by immunoblotting shown in (Fig 5A and 5B). HIF1*A*-deficient cells had significant decrease in DENV2 RNA levels in normoxia (21% $O_2$) (Fig 5C), as compared to control (cells were transfected with control-shRNA). Further, the transcripts of IFNA1 and IFNB1 (Fig 5D and 5E) and also IRF3, IRF7 and IRF9 (Fig 5F – 5H) were elevated in HIF1*A* -deficient cells as compared to controls.

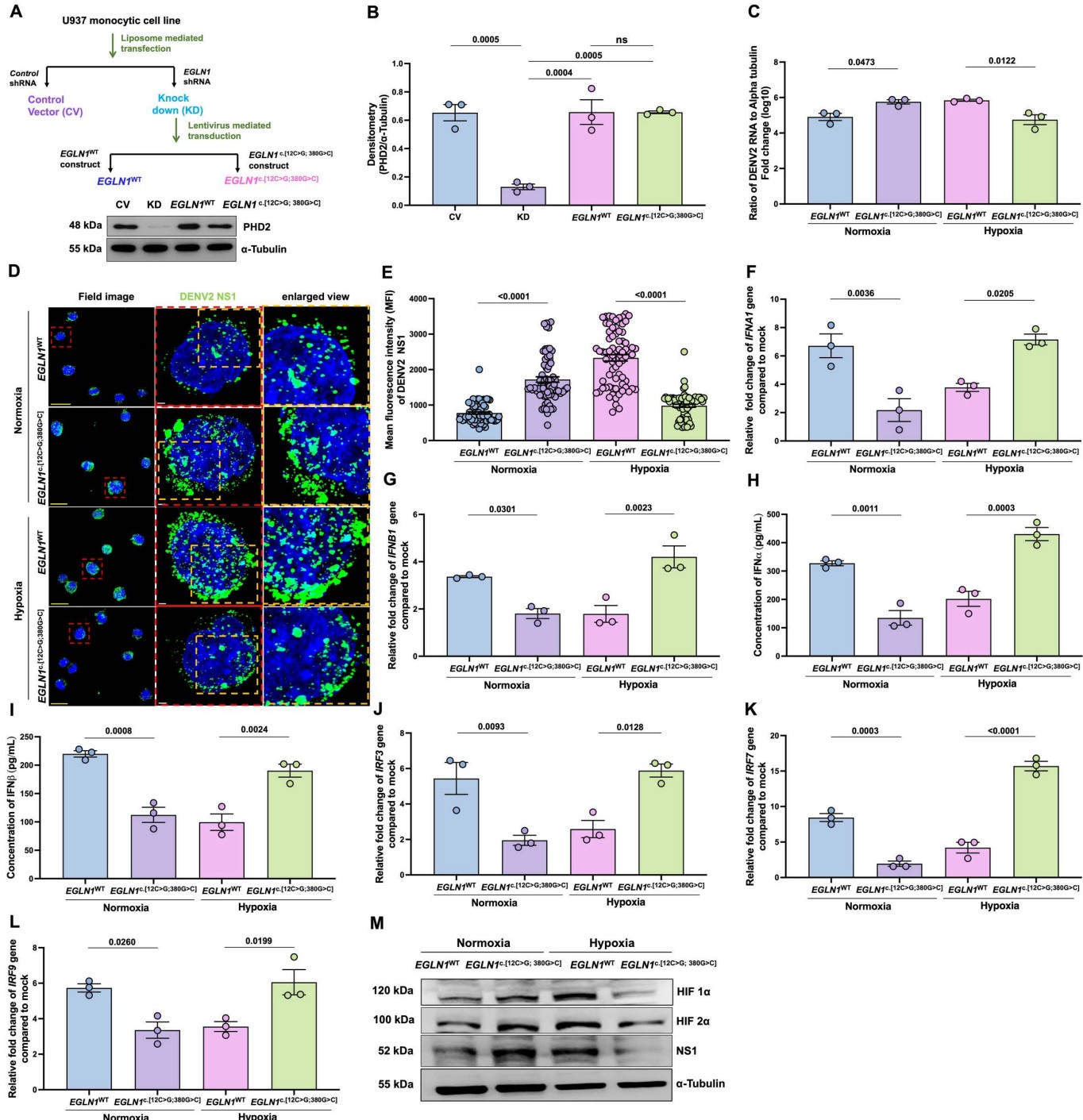

**Fig 2. DENV2 infection in U937 cell line engineered with EGLN1c.[12C>G; 380G>C] or EGLN1WT under normoxia or hypoxia.** A-B. Endogenous PHD2 level was knocked down using shRNA targeting 3'UTR of EGLN1, and EGLN1WT(PHD2WT) or EGLN1c.[12C>G; 380G>C] (PHD2D4E;C127S) construct was introduced in U937 monocytic cells. Densitometry data were normalized with α-Tubulin. one-way ANOVA and Bonferroni's post-test were used for analysis. Cells were infected with DENV2 and exposed to normoxia or hypoxia, as mentioned in Fig 1. **C.** DENV2 RNA levels were measured using qRT-PCR, **D.** DENV2 NS1 (green) and nucleus (DAPI, blue) were measured in the above cells and quantifies as **E.** MFI. Data are mean ± SEM from 3 independent experiments, one-way ANOVA and Bonferroni's post-test were used for statistical analysis. **F.** IFNA1 and **G.** IFNB1 levels were measured using qRT-PCR, relative fold-change after normalization with human α-Tubulin, as compared to mock. One-way ANOVA and Bonferroni's post-test were used

for analysis. **H.** IFNα and **I.** IFNβ concentrations in cell supernatant were measured using ELISA and calculated as mentioned above. **J.** Relative fold change of IRF3, **K.** IRF7 and **L.** IRF9 genes were measured from cell pellets and calculated as mentioned above. **M.** Expression of HIF1α, HIF2α and Dengue NS1 proteins were measured using western blot. Densitometry data were normalized with α-Tubulin (S2A–S2C Fig). Data are mean±SEM from 3 independent experiments and P-values are mentioned in graphs.

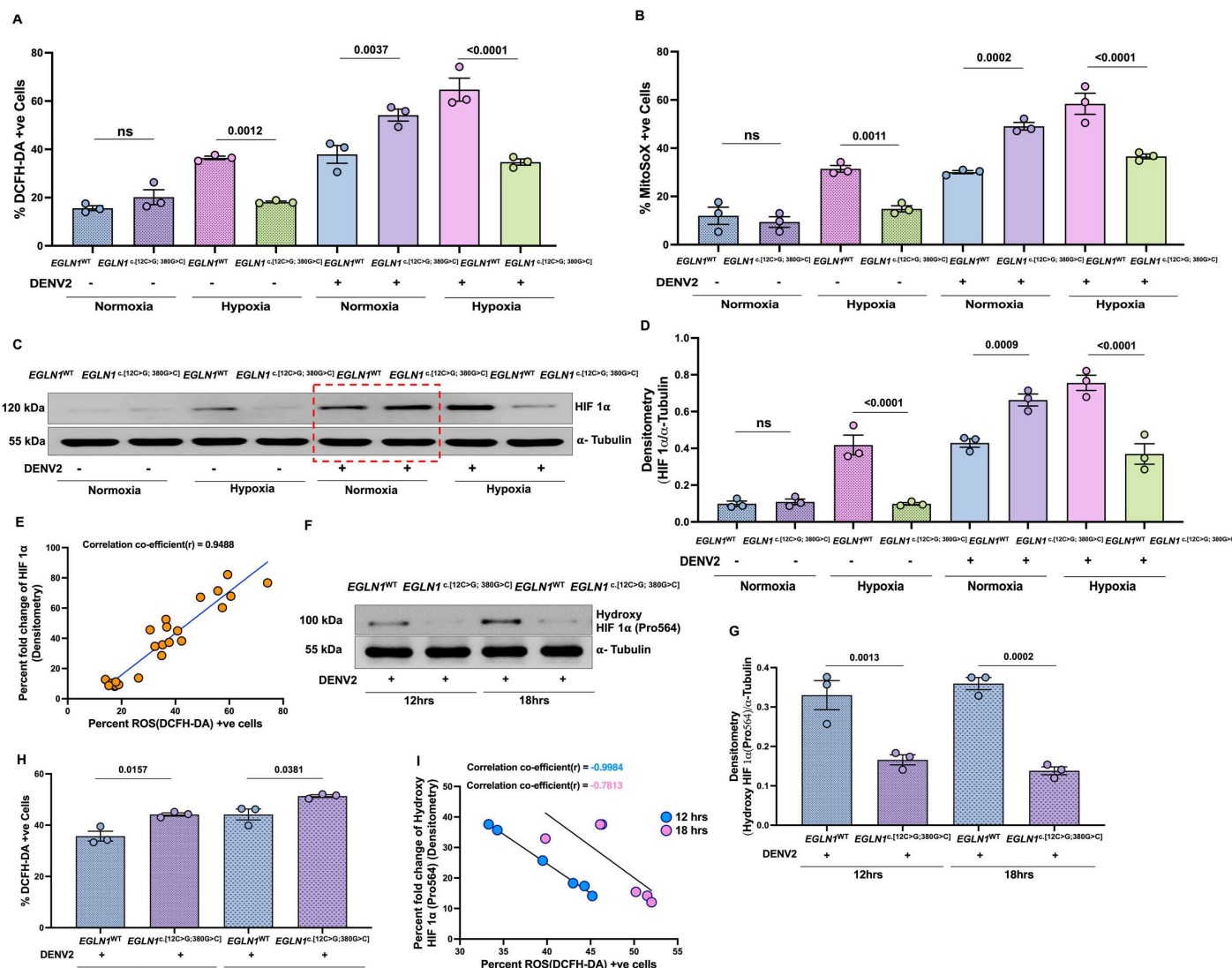

**Fig 3. ROS generation in DENV2 infected U937 cell lines engineered with *EGLN1*<sup>c.[12C>G; 380G>C]</sup> or *EGLN1*<sup>WT</sup> under normoxia or hypoxia.** U937 monocytic cell lines expressing *EGLN1*<sup>c.[12C>G; 380G>C]</sup> or *EGLN1*<sup>WT</sup> were exposed under normoxia (21% $O_2$) or hypoxia (3% $O_2$) for 8 hrs. Followed by cells were infected with DENV2 (MOI~3) and incubated for another 16 hrs. Cells were collected and **A.** Cellular total ROS and, B. mitochondrial ROS were quantified using DCFH-DA and Mito-SOX dye respectively. C-D. HIF1α expression was measured from above cell pallets using western blot. Densitometry data were normalized with α-Tubulin. **E.** Correlation coefficient was measured between HIF1α expression and ROS (DCFH-DA) positive cells, and a positive correlation was observed, Pearson's correlation **(r)** ~ 0.9488. In continuation of experiment highlighted in Fig 3C, we measured Hydroxy HIF1α(Pro564) in these cells expressing *EGLN1*<sup>c.[12C>G; 380G>C]</sup> or *EGLN1*<sup>WT</sup> at 12 and 18 hrs after DENV2 infection under normoxia (21% $O_2$). F-G. Expression of Hydroxy HIF1α(Pro564) protein was measured using western blot. Densitometry data were normalized with α-Tubulin, calculated from 3 independent experiments. Data are mean±SEM. **H.** Cellular total ROS was quantified using DCFH-DA dye, and represented as mentioned above. Data are mean±SEM from 3 independent experiments, one-way ANOVA and Bonferroni's post-test were used for statistical analysis. **I.** Correlation coefficient between Hydroxy HIF1α(Pro564) expression and ROS(DCFH-DA) positive cells, and a negative correlation were observed, r~-0.9984 and -0.7813 at 12 and 18 hrs respectively. Gating strategy is mentioned in S1B – S1C Fig.

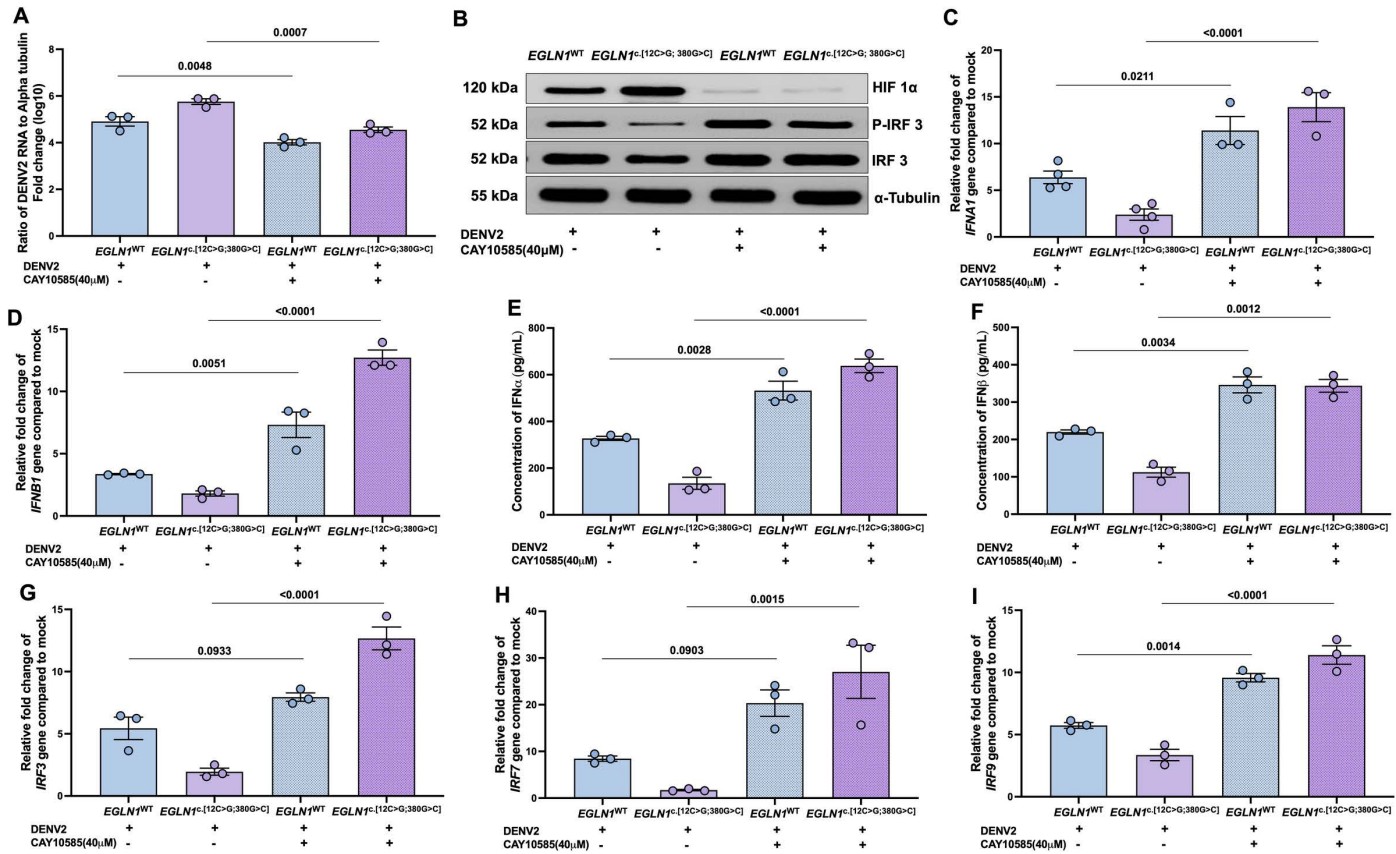

**Fig 4. DENV2 infection and IFN synthesis in EGLN1^c.[12C>G; 380G>C] or EGLN1^WT U937 cells in presence of HIFα inhibitor.** EGLN1^c.[12C>G; 380G>C] or EGLN1^WT expressing U937 cells were infected with DENV2 as mention in Fig 2, in presence of HIFα inhibitor (CAY10585, 40μM). **A.** DENV2 RNA levels were measured and calculated as mentioned above. **B.** Expression of HIF1α, phosphorylated(P)-IRF-3, IRF3 were measured using western blot. Densitometry data from 3 independent experiments (S2D–S2F Fig). **C.** IFNA1 and **D.** IFNB1 levels from cell pellets were measured using qRT-PCR, relative fold-change after normalization with human α-Tubulin. One-way ANOVA and Bonferroni's post-test were used for analysis. **E.** IFNα and **F.** IFNβ concentrations in cell supernatant were measured using ELISA. One-way ANOVA and Bonferroni's post-test were used for analysis. **G.** IRF3, **H.** IRF7 and **I.** IRF9 genes were measured from the above cell pellets and data were calculated as mentioned above. Data are mean±SEM from 3 independent experiments, and P-values are mentioned in graphs.

## PHD2^D4E;C127S Tibetans previously exposed to COVID-19 infection/vaccine have elevated IFNγ expression in T cells upon exposure to SARS-CoV-2 RBD peptides in hypoxia

We further examined the role of PHD2^D4E;C127S variant in COVID-19 infection. We used PBMCs isolated from PHD2^D4E;C127S or PHD2^WT Tibetans, were previously infected with COVID-19. We incubated these PBMCs with SARS-CoV-2 RBD peptide under normoxia (21% $O_2$) and hypoxia (3% $O_2$). We found a significant overexpression of IFNγ in both CD4 (Fig 6A) and CD8 (Fig 6B) T cells of PHD2^D4E;C127S individuals under hypoxia. Other hand, a lesser expression of IFNγ in PHD2^D4E;C127S T cells in normoxia (21% $O_2$) as compared to PHD2^WT cells.

## Discussion

Our study, for the first time, describes the unique crosstalk between the Tibetan-specific PHD2^D4E;C127S variant and hypoxic environmental variations in the regulation of host innate/adaptive immune responses against dengue and COVID-19 infections. The evolutionarily selected haplotype c.[12C > G; 380G > C] in *EGLN1* has about ~85% gene

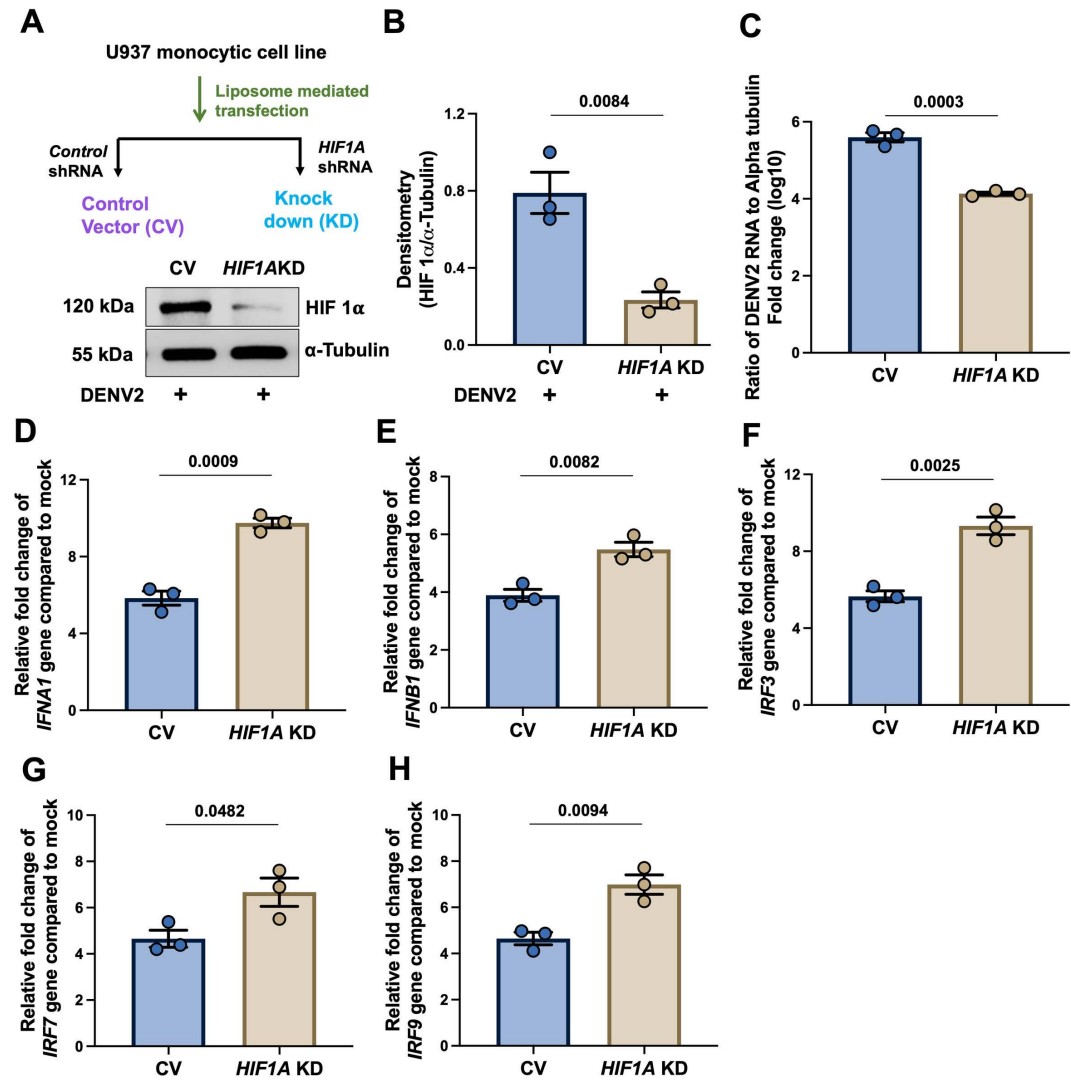

**Fig 5. DENV2 infection and IFN expression in HIF1 α-knock down U937 cells. A-B.** HIF1α-knockdown U937 cells were generated using HIF1A shRNA. Control shRNA was used. Densitometry data were normalized with α-Tubulin. Unpaired t test was used for analysis. Cells were infected with DENV2 and exposed to hypoxia as described in Fig 2. **C.** DENV2 RNA levels were measured. Unpaired t test was used for analysis. Similarly, **D.** IFN*A1*, **E.** IFN*B1*, **F.** IRF3, **G.** IRF7 and **H.** IRF9 genes were measured using qRT-PCR from above cell pellets and data were calculated as described above, relative fold-change after normalization with human α-Tubulin, as compared to mock. Data are mean±SEM from 3 independent experiments, and P-values are mentioned in graphs.

frequency in Tibetans as is not found among other ethnic populations. Having a high affinity for $O_2$, PHD2[D4E;C127S] destabilizes HIF-1/2α at low $pO_2$ of a hypoxic environment, whereas PHD2[WT] fails to inactivate the HIFs in this condition. Previously, we reported that the Tibetans carrying the gain-of-function PHD2[D4E;C127S] variant are protected from the hypoxia-induced maladies like polycythemia [11], and inflammatory events like pulmonary/brain edema [20] at the high altitudes. Monocytes from PHD2[D4E;C127S] Tibetans had significantly lower DENV2 infection under hypoxia compared to PHD2[WT] monocytes, who had higher infection load in hypoxia in vitro. Interestingly, the PHD2[D4E;C127S] monocytes showed a high DENV2 infection in normoxia. A suppressed expression of type-1 interferon (IFN-1, IFNα and IFNβ) was observed in these mutant cells. Elevated ROS suppressed hydroxylation of HIF1α by PHD2[D4E;C127S] in

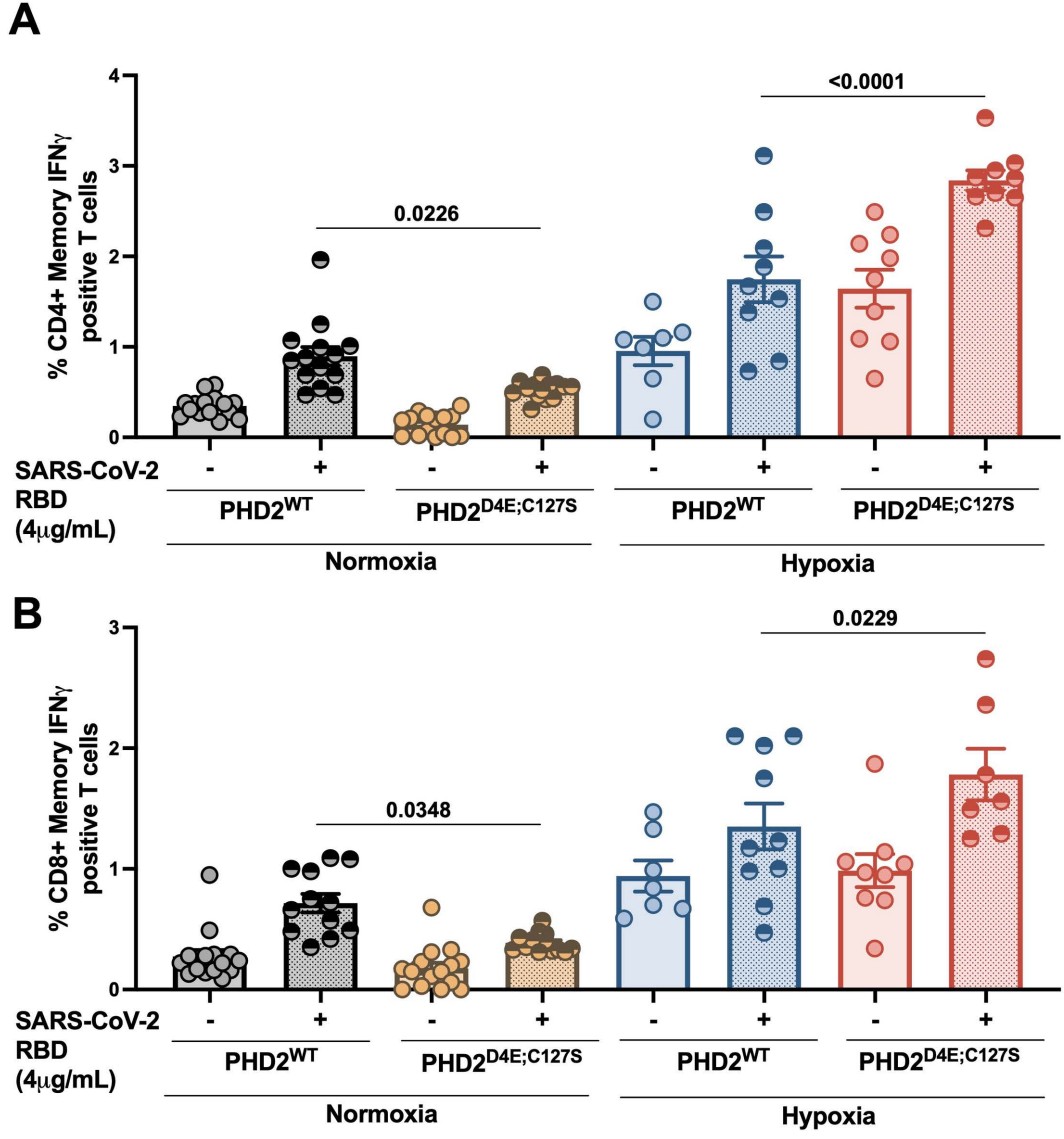

**Fig 6. IFNγ expression in PHD2<sup>WT</sup> or PHD2<sup>D4E;C127S</sup> CD4/CD8 T cells from Tibetans upon exposure to SARS-CoV-2 RBD peptide under hypoxia or Normoxia.** PBMCs were collected from PHD2<sup>WT</sup> or PHD2<sup>D4E;C127S</sup> Tibetans who were either pre-exposed to SARS-CoV-2 infection and/or vaccinated with SARS-CoV-2 vaccines, and incubated with purified SARS-CoV-2 RBD peptide (4 µg/mL of peptide) and incubated under normoxia (21% $O_2$) or hypoxia (3% $O_2$) for 24 hrs. A. CD4 + Memory IFNγ positive T cells and, B. CD8 + Memory IFNγ positive T cells were measured using flow cytometry. Gating strategy is mentioned in S1F Fig, One-way ANOVA and Bonferroni's post-test were used for analysis, Data are mean ± SEM percentage of cells, and P-values are mentioned in the graphs.

infected monocytes, in turn increasing the stabilization of HIF1α in normoxia. The ROS-mediated inhibition of PHD2 and stabilization of HIF1α was previously reported by others [18,19]. However, the mechanism of elevation of ROS in monocytes in PHD2<sup>D4E;C127S</sup> cells upon viral infection in normoxia is still unclear, and a further studies are needed. We now describe that the suppressed PHD2<sup>D4E;C127S</sup>, and resulting elevated HIF1α, and decreased the IFN synthesis in these mutant cells in normoxia upon infection. The suppressive effect of HIF1α on IFN-1 (IFNα and IFNβ) or IFN-2 (IFNγ) was described by others in a hypoxic microenvironment in cancer [21–25] as well as in SARS-CoV-2 infection

[26]. It has also been described that treatment with HIF1α-inhibitor significantly improved IFN-α/β synthesis, in turn decreasing viral infections in *EGLN1*[c.[12C>G;380G>C]] (PHD2[D4E;C127S]) cells in vitro. Other studies have also described that HIF1α-inhibitor alleviates immunosuppression by enhancing IFNγ expression [27].

Our study also elucidates the role of PHD2[D4E;C127S] variant in protection against COVID-19 infection. The PBMC collected from Tibetan individuals, exposed to prior SARS-CoV-2 infection or received the related COVID-19 vaccines, when incubated with SARS-CoV-2 RBD peptide, had elevated expression of IFNγ in PHD2[D4E;C127S] memory T cells under hypoxia, indicating a protective effect of these variants against COVID-19 infections in native Tibetan highlanders living under a hypobaric hypoxic environment. The lower prevalence of COVID-19 among native highlanders than lowlanders has also been reported [7,8]. Besides, our study also describes a lesser expression of IFNγ in PHD2[D4E;C127S] T cells under normoxia, indicating a higher chance of infection with COVID-19 and a similar type of response by these mutant cells against dengue.

In summary, we describe that 1) the evolutionarily selected PHD2[D4E;C127S] variant correlates with the decreased viral infection in hypoxia, but higher infection in normoxia. 2) Conversely, PHD2[WT] has opposite trends. 3) HIF1α-IFN axis as the potent therapeutic target to protect from viral infections, depicted in schematic Fig 7. 4) HIF1α inhibitor improves IFNα/β as well as IFNγ synthesis, in turn providing protection against viral infections.

## Limitations of the study

Caveats of our study include the association of the PHD2[D4E;C127S] variant with real-time infection with either dengue or COVID-19 in Tibetan individuals at high altitude and sea level. Secondly, investigating the mechanism of elevated ROS generation, in turn suppression of PHD2[D4E;C127S] and stabilization of HIF1α, and a decrease in IFN synthesis, inversely correlating with increased in viral infection in PHD2[D4E;C127S] monocytes as compared to PHD2[WT] in normoxia.

## Materials and methods

### Ethics statement

Human ethics approval was obtained from the Institutional Ethics Committee (IEC) of the Regional Centre for Biotechnology (RCB) to study the human samples (ref. no. RCB-IEC-H-01 and RCB-IEC-H-22). The written consent form was collected from all the volunteers who donated blood.

### Study subjects

We collected 5 mL of whole blood in citrate dextrose anticoagulant vials from healthy Tibetan volunteers from Leh, India (>3600 m altitude) and Delhi, India (sea level) of any gender, aged between 18–45 years without having any history of chronic diseases including diabetes, cancer, kidney failure, autoimmune disease, no chronic drug use, (details are mentioned in S1 Table). Peripheral blood mononuclear cells (PBMC) were used for further studies.

### Genotyping of SNPs

We isolated genomic DNA from whole blood using a Qiagen FlexiGene DNA kit. From genomic DNA, the genotyping of the samples was performed. Our targeted SNPs of *EGLN1* (NCBI RefSeq NG_015865) are present in exon 1 which were amplified by PCR using the specific primers X1F (CCCCTATCTCTCTCCCCG) and X1R (CCTGTCCAGCACAAACCC). Phusion Polymerase Master Mix (GC rich) was used. Amplification was done at 98°C for 5 mins, 98°C for 120 secs (40 cycles), 60.1°C for 60 secs, and 72°C for 120 secs, followed by final extension at 72°C for 5 mins. It generated a product size of 1025bps. The amplified product was then run in 1% agarose gel and then purified using a Qiagen gel extraction kit and sequenced to identify the SNPs c.12C>G; 380G>C [20].

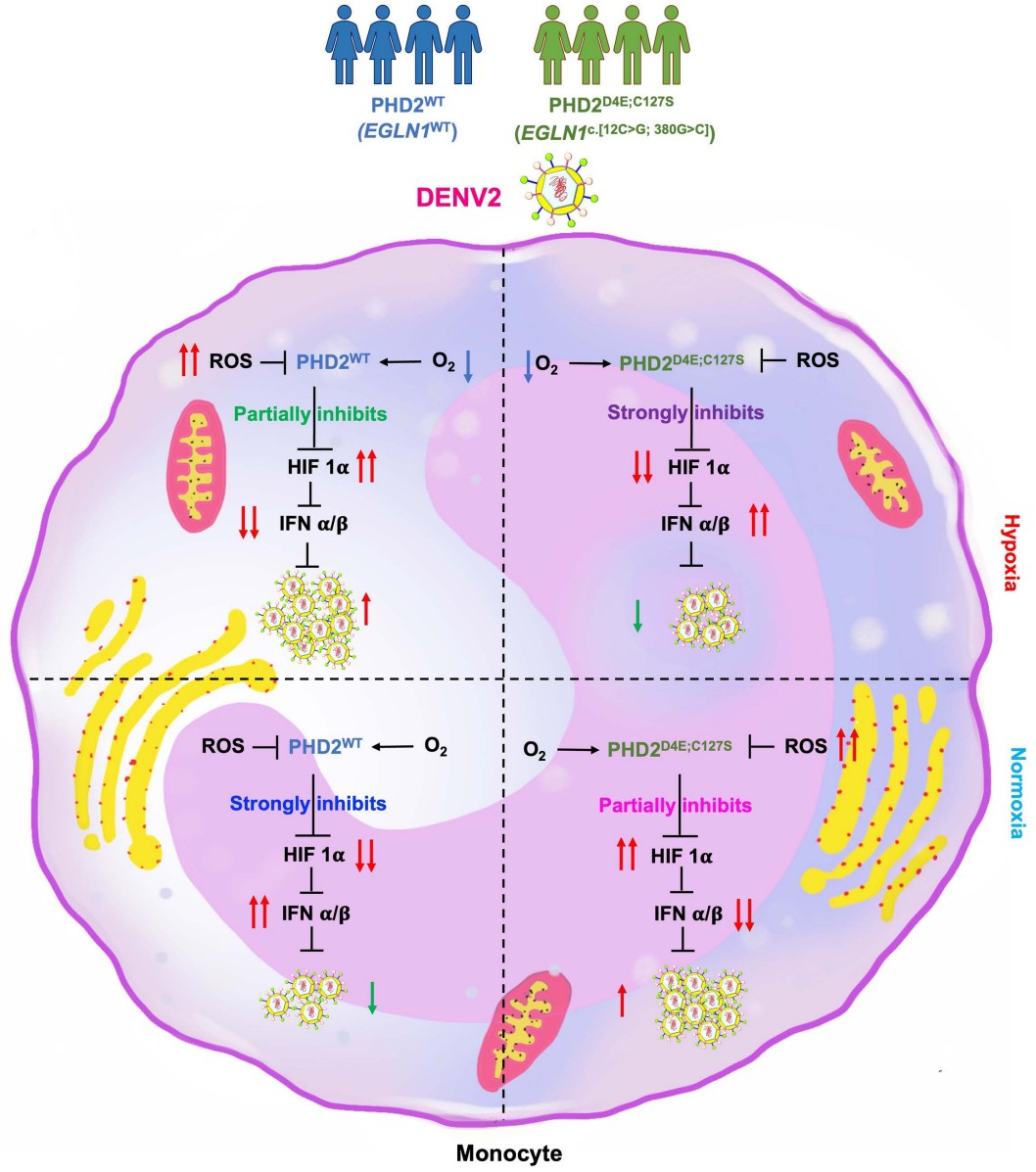

**Fig 7. The crosstalk between the Tibetan PHD2<sup>D4E;C127S</sup> variant with pO$_2$ in regulating HIF:IFN axis in monocytes in dengue and COVID-19 infections.** The observation in Tibetans serves as a model for us to appreciate the crucial role this gain-of-function mutation play in modulation of HIF1α-mediated IFN responses against viral infections. Hereafter, we attempted to resolve HIF1α-associated viral pathogenesis in vitro by augmenting the synthesis of IFNs using a HIF1α inhibitor.

## PBMC isolation

Whole blood was centrifuged at 500 rpm for 15 mins at room temperature for isolating Platelet Rich Plasma (PRP) and leukocytes. PRP was centrifuged to separate platelet and plasma [Platelet Poor Plasma (PPP)] at 1000 g for 15 minutes and the bottom red fraction layer was diluted with 1XPBS at a 1:1 ratio and gently pipetted on top of each 3 mL of Lymphoprep (Serumwerk), followed by a centrifugation at 550 g with acceleration 5 and break free for 26 mins at room temperature. The middle cloudy layer containing Peripheral Blood Mononuclear Cells (PBMCs) was collected and washed with 1XPBS twice.

## Monocyte isolation

After isolating PBMCs from whole blood by density gradient centrifugation, PBMCs were incubated with CD14$^+$ micro-bead (Miltenyibiotec, Germany) and kept for 45 mins at 4°C. The monocytes were isolated through MACS column via positive selection. Isolated monocytes were seeded ($0.35 \times 10^6$ cells/well) in 12-well cell plates (Corning, NY, USA), in RPMI-1640 media (Sigma Aldrich, St. Louis, USA) and supplemented with 10% (v/v) heat-inactivated fetal bovine serum (FBS; Gibco Invitrogen, USA), 1% (v/v) penicillin-streptomycin (Sigma Aldrich, USA) for 2 hrs at 37 °C in 5% $CO_2$ incubator and used for further experiments.

## Cells and Viruses

The human monocytic cell line U937 and monkey kidney cell line Vero E6 cells (all from ATCC, USA) were cultured at 37°C under 5% $CO_2$ and 70–80% humidity in either RPMI or DMEM (Invitrogen and Sigma Aldrich) along with 10% heat inactivated FBS and 1% (v/v) penicillin-streptomycin. The original cell lines were tested for mycoplasma contamination-free. The viruses Dengue virus subtype-2 (DENV2; NGC strain) was used for experiments at MOI~3 *in vitro*.

## Virus preparation

DENV2 NGC strain was propagated in mosquito cell line C6/36. This cell line was cultured in L15 media supplemented with 10% (v/v) heat-inactivated FBS and 1% (v/v) penicillin-streptomycin-glutamine. DENV2 infection was given to cells at MOI~0.1 for 2 hrs at 37°C, 5% $CO_2$ under continuous gentle rocking conditions. At the time of infection cells were supplemented with L15 media having only 2% FBS. After removal of the infection, cells were cultured in complete L15 media having 10% FBS and kept for 5 days at 37 °C, 5% $CO_2$. Cell supernatant was then collected, centrifuged at 1200 rpm for 15 mins and purified through 100 kDa cut-off membrane before use. This purified cell supernatant was aliquoted and stored at -80°C for further use. The amounts of infectious particles were expressed as focus-forming units (FFU)/mL and titrated in Vero cells, details are mentioned below.

## Focus-forming assay/Plaque forming assay

DENV2 virus stock was diluted serially and infected in Vero cells for 2 hrs. After 2 hrs, the infection was then removed and infected cells were supplemented with fresh media at 37°C and 5% $CO_2$ for 3 days. Cells were then fixed with 2% paraformaldehyde followed by permeabilization with 0.1% Triton X-100 (in 1X PBS) for 20 mins and blocking with 5% goat serum (in PBS) for 1 hr. Primary anti-mouse 4G2 antibody was then added to cells for overnight at 4°C, followed by addition of goat anti-mouse AF488-conjugated secondary antibody for 2hrs at room temperature. Cells were washed with 1XPBS twice. Foci were visualized in a fluorescence microscopy and counted to evaluate the virus titre.

## Generation of monocytic cell line expressing *EGLN1*$^{WT}$ (PHD2$^{WT}$) or *EGLN1*$^{c.[12C>G; 380G>C]}$ (PHD2$^{D4E;C127S}$) and *HIF1A* knocked down cell lines

In U937 monocytic cell line, the endogenous PHD2 expression was depleted using liposome-mediated delivery (Thermo Fisher Scientific, USA) according to the manufacture's protocol using shRNA targeting 3' Untranslated region. Opti-MEM media was used during transfection. After anti-biotic selection, knock down in cells was confirmed using Western blot assay. In knock down cells, *EGLN1*$^{WT}$ (PHD2$^{WT}$) or *EGLN1*$^{c.[12C>G; 380G>C]}$ (PHD2$^{D4E;C127S}$) construct were introduced using lentivirus particles. 4µg/mL polybrene was added to media during transduction followed by centrifugation at 3000 rpm, at 32°C for 90 mins. Cells were selected using media containing specific anti-biotic and confirmed using western blot assay. Details are mentioned in Fig 2A and 2B.

In the U937 monocytic cell line, the endogenous HIF1α expression was knocked down using liposome-mediated delivery (Thermo Fisher Scientific, USA) according to the manufacture's protocol. Opti-MEM media was used during

transfection, followed by anti-biotic selection and knock down in cells was confirmed using Western blot assay. Details are mentioned in Fig 5A and 5B, and described in our previous work [20].

### In vitro experimental design

Primary monocytes/U937 cells (expressing $EGLN1^{WT}$ (PHD2$^{WT}$) or $EGLN1^{c.[12C>G;\ 380G>C]}$ (PHD2$^{D4E;C127S}$)/ HIF1α-knock down U937 cells were kept under normoxia (21% $O_2$) and hypoxia (3% $O_2$). Cells were incubated under hypoxia for pre-exposure to mimic the high altitude environment for 8 hrs then were infected with DENV2 at MOI~3 for 2 hrs and kept for another 14 hrs for viral multiplication. Virus inoculum was prepared in media containing 2% heat-inactivated FBS. After 2 hrs, infections were removed and cells were supplemented with complete media containing 10% FBS. Cells and supernatants were collected for further analysis.

U937 cells [(expressing $EGLN1^{WT}$ (PHD2$^{WT}$) or $EGLN1^{c.[12C>G;380G>C]}$ (PHD2$^{D4E;C127S}$)] were treated with CAY10585 (HIFα-inhibitor, CAYMAN chemical) at 40μM concentrations for 8 hrs before infection and also kept 14 hrs post-infection.

### Real-Time PCR

Total RNA was isolated from the primary monocytes/U937 cells expressing $EGLN1^{WT}$ (PHD2$^{WT}$) or $EGLN1^{c.[12C>G;\ 380G>C]}$ (PHD2$^{D4E;C127S}$) construct)/ HIF1α-KD U937 cell pellet using RNAiso Plus reagent (Takara Bio, Japan). In detailed, 0.2 mL of chloroform was added per mL RNAiso Plus reagent used for each sample followed by vigorous mixing for 15secs. After incubation of 3–5 mins at room temperature, samples were centrifuged at 12,000g for 15 mins at 4°C. Upper aqueous phase was collected into new fresh tubes and 0.5mL of 100% isopropanol was added into it per mL RNAiso Plus reagent used followed by 20 mins of incubation at room temperature. Centrifugation was performed at 12,000g for 10 mins at 4°C. Supernatant was removed and washed the pellet with 1mL of 75% ethanol per mL of RNAiso Plus reagent used. Samples were vortexes and centrifuged at 7500g for 5 mins at 4°C followed by the pellet was air dried and resuspended in RNase free water and incubated at 55–60°C for 10–15 mins, concentration and purity were measured using nanodrop 2000 (Thermo Scientific) machine. Stored at -80°C for further use.

DNA was prepared from 1000 ng of RNA by iScript reverse transcriptase, primers from Super Script IV First-strand synthesis system (BIORAD, USA). 2μL of cDNA from each set was used for qPCR using forward primer 5'3' and reverse primer 5'3' with SYBR green on Quant Studio 6 Flex. List of primers are mentioned in S2 Table. Data were calculated using δCT value using Threshold cycle (CT) value of α-Tubulin/ GAPDH as the housekeeping control. Relative gene expression was calculated using 2^- δCT.

### Confocal microscopy

After completion of experiments, cells were fixed with 4% paraformaldehyde for 20 mins at room temperature (RT) under normoxia or hypoxia followed by permeabilizing with 0.1% Triton X-100 for 20 mins, and blocked in 5% goat serum for 1 hr at RT. Primary antibodies were probed and kept overnight at 4°C. Cells were washed twice with 1X PBS and then labelled with Alexa-488/594 conjugated secondary antibody for 2hrs at room temperature. After the incubation was over, cells were again washed with 4–5 times with 1X PBS to remove non-specific binding. Cells were then stained with DAPI for 20 mins at room temperature followed by mounting with mounting media (Prolong Gold). Images were taken in a Leica Confocal DMI 6000 TCS-SP8 microscope (Leica Microsystems, Wetzlar, Germany) at 63 × oil immersion objective (NA 1.4) Plan Apo objectives and quantified. Images were taken using Z-stacks at 0.50 μm per slice by sequential scanning. For image processing, Image J Fiji software was used. In detailed, first cross-sectional and maximum intensity projection images were generated followed by integrity density of the images were calculated and deducted the background density from it. Background density was measured from the area without fluorescence. Total fluorescence intensities were measured and calculated from minimum 15 images in each set with three individual experiments.

## Flow cytometry

After completion of the experiment, the infected primary monocytes, were first stained using anti-CD14 and anti-CD11c (surface staining) for 45 mins at room temperature in slow rocking condition, and then processed for intracellular staining. The cells were washed with 1X PBS and then fixed with BD fixation buffer for 20mins at room temperature followed by permeabilization with a permeabilization buffer for 30mins at room temperature, and then labelled with anti-human HIF1α for overnight at 4°C. Cells were washed twice with 1X PBS, followed by incubation with goat anti-rabbit AF488 secondary antibody for 2 hrs at room temperature. The cells were washed twice with 1X PBS and processed in a FACS Verse (BD Biosciences, San Jose, USA). Data were analysed using FlowJo software (Treestar, Ashland, USA). Detailed gating strategy is mentioned in S1A Fig.

PBMCs from Tibetan individuals were treated with or without SARS-CoV-2 RBD peptide and kept under normoxia (21% $O_2$) and hypoxia (3% $O_2$) for 16 hrs. Cells were first stained with CD3 APC, CD4 V450, CD8 PerCP, CD45RO PE-CY-7 followed by fixation with fixative for 20mins at room temperature. Further, cells were permeabilized using permeabilization buffer for 30mins at room temperature followed by intracellular IFNγ PE staining was done. Cells were washed twice with 1X PBS and acquired using a FACS Verse (BD Biosciences). Data were analysed as mentioned above.

## ELISA

The human IFNα and IFNβ were measured in cell supernatants using a commercially available ELISA kits (Invitrogen and R&D systems) as per manufacturer's protocol.

## Immunoblotting

The intracellular proteins PHD2, HIF1α, Hydroxylated HIF1α (Pro564), P-IRF3, IRF3, α-Tubulin (Cell Signalling Technology), HIF2α, Dengue NS1 and VEGF (Abcam) were detected in primary monocytes, U937 monocytic cell line. Cell pellets were lysed using RIPA along with protease-phosphatase inhibitor (Thermo Scientific Life Tech). SDS-PAGE gel was run followed by immunoblotting using mentioned primary antibodies and kept overnight at 4°C. Blots were washed with 0.1% Tween-20 three times 10 mins each followed by HRP-conjugated goat anti-mouse/rabbit secondary antibody for 2 hrs at room temperature. Blots were washed again 5–6 times after secondary antibody incubation is done and images were captured in ImageQuant LAS 4000 using Immobilon Forte Western HRP Substrate. All the antibodies are commercially validated and cited in published literature. List of all antibodies were mentioned in S3 Table.

## Total ROS and Mitochondrial ROS measurement

After completion of in vitro experiments, cells were washed with 1X PBS followed by staining with either DCFH-DA (50μM), cell-permeable intracellular ROS probe (Sigma-Aldrich, USA) or MitoSOX (5μM), mitochondrial ROS indicator (Invitrogen, USA) at 37°C for 30 mins at dark in gentle rocking condition. After that, Cells were washed twice with 1X PBS and acquired in a FACS Verse (BD Biosciences, San Jose, USA). Data were analysed using FlowJo software (Treestar, Ashland, USA).

## Statistical analysis

The Experimental values were calculated from at least three independent experiments and presented as mean ± SEM (Standard error of the mean). Statistical analysis was performed by either an unpaired t-test or One-way ANOVA followed by Bonferroni post-test if data was distributed normally, paired t-test or Kruskal Wallis test for non-normal distributed data. Graph pad Prism 9.5.0 software was used to analyse the data and p-values <0.05 were considered as significant.

## Supporting information

**S1 Fig. Flow cytometry gating strategy.** A. Gating strategy for HIF1a positive primary monocytes. DENV2 infected PBMCs from PHD2$^{WT}$ and PHD2$^{D4E;C127S}$ individuals, were stained with CD14-PE and CD11c-V450, and intracellular HIF1a (using primary and goat anti-rabbit AF488 secondary antibodies). B-C. Gating strategy for DCFH-DA (total ROS measurement) and MitoSOX (mitochondrial ROS measurement) positive cells. D. IFNγ positive CD4$^+$ and, E. CD8$^+$ Memory T cells in Non- Tibetan individuals were measured. Unpaired t test was used for analysis. Each dot represents a single individual. Data are mean±SEM, P values were mentioned. F. Gating strategy for IFNγ$^+$ T cells. PBMCs from PHD2$^{WT}$ and PHD2$^{D4E;C127S}$ individuals were stained for CD3 APC, CD4 V450, CD8 PerCP, CD45RO PE-Cy-7 and intracellular IFNγ PE markers.
(TIFF)

**S2 Fig. Densitometry of western blots.** Densitometry analysis of the protein bands were analysed using the Image-J software; main Fig 2M (S2A – S2C Fig; one-way ANOVA followed by Bonferroni's post-test was used), Fig 4B (S2D – S2F Fig; one-way ANOVA followed by Bonferroni's post-test was used), Data are mean±SEM from triplicate blots. P-values are mentioned in the graphs.
(TIF)

**S3 Fig. Expression of VEGF and cell death assay.** Expression of HIF target molecule, A.VEGF was measured through western blot in *EGLN1*$^{c.[12C>G; 380G>C]}$ or *EGLN1*$^{WT}$ expressing U937 cells infected with DENV2 as mentioned in Fig 4, in presence of HIFα inhibitor (CAY10585, 40µM). B. Densitometry analysis from 3 independent experiments of the protein band of VEGF, normalized with α-Tubulin. Data are mean±SEM, one-way ANOVA followed by Bonferroni's post-test was used. C. Cell death assay using Propidium iodide (100µg/µL) for staining *EGLN1*$^{c.[12C>G; 380G>C]}$ or *EGLN1*$^{WT}$ expressing U937 cells infected with DENV2 under normoxia and hypoxic conditions. Data are mean±SEM from 3 independent experiments. One-way ANOVA and Bonferroni's post-test were used for analysis. P-values are mentioned in the graphs.
(TIF)

**S4 Fig. Pro-inflammatory cytokines in DENV2 infected cells.** A-C. Pro-inflammatory cytokines *TNFA*, *IL6* and *IL1B* were measured using qRT-PCR from *EGLN1*$^{WT}$(PHD2$^{WT}$) or *EGLN1*$^{c.[12C>G; 380G>C]}$ (PHD2$^{D4E;C127S}$) experimental cell pellet (infected with DENV2) mentioned in Fig 2. Relative fold-change after normalization with human α-Tubulin. Data are mean±SEM, One-way ANOVA and Bonferroni's post-test were used for analysis. D–F. Proinflammatory cytokines TNFα, IL6 and IL1β protein levels were measured using CBA from above cell supernatants. Data are mean±SEM, one-way ANOVA and Bonferroni's post-test were used. P values are mentioned in the graphs.
(TIF)

**S5 Fig. Sequencing of *EGLN1*$^{WT}$(PHD2$^{WT}$) or *EGLN1*$^{c. [12C>G; 380G>C]}$ (PHD2$^{D4E;C127S}$) in U937 cells.** Sequencing/Genotyping of *EGLN1*$^{WT}$ or *EGLN1*$^{c.[12C>G; 380G>C]}$ in A-B. U937 cells.
(TIF)

**S6 Fig. DENV2 RNA levels in *EGLN1*$^{WT}$(PHD2$^{WT}$) or *EGLN1*$^{c.[12C>G; 380G>C]}$ (PHD2$^{D4E;C127S}$) U937 cells infected with DENV2 for 12hrs and 18hrs under normoxia.** U937 monocytic cell lines expressing *EGLN1*$^{c.[12C>G; 380G>C]}$ or *EGLN1*$^{WT}$ were infected with DENV2 (MOI~3) and incubated for 12hrs and 18hrs under normoxia (21% O$_2$). Cells were collected and A. DENV2 RNA levels were measured using qRT-PCR, Data are mean±SEM from 3 independent experiments, one-way ANOVA and Bonferroni's post-test were used for statistical analysis.
(TIF)

**S7 Fig. Striking image.**
(TIF)

**S1 Table. Study subjects.**
(TIFF)

**S2 Table. List of primers used in this study.**
(TIF)

**S3 Table. List of antibodies used in this study.**
(TIF)

**S4 Table. List of reagents used in this study.**
(TIF)

**S1 File. The raw data of all main figures are mentioned in the xlsx file.**
(XLSX)

**S2 File.** The raw data of all supplementary figures are mentioned in the xlsx file.
(XLSX)

**S3 File. The raw data of all western blots are mentioned in the pdf file.**
(PDF)

## Author contributions

**Conceptualization:** Josef T Prchal, Prasenjit Guchhait.

**Data curation:** Riya Ghosh, Garima Joshi, Nishith M Shrimali, Tsewang Chorol, Tashi Thinlas, Parvaiz A Koul.

**Formal analysis:** Riya Ghosh.

**Investigation:** Riya Ghosh, Garima Joshi, Nishith M Shrimali, Kanchan Bhardwaj.

**Methodology:** Riya Ghosh, Garima Joshi, Nishith M Shrimali, Kanchan Bhardwaj, Tsewang Chorol, Tashi Thinlas, Parvaiz A Koul.

**Project administration:** Prasenjit Guchhait.

**Resources:** Prasenjit Guchhait.

**Supervision:** Prasenjit Guchhait.

**Validation:** Prasenjit Guchhait.

**Writing – original draft:** Riya Ghosh, Nishith M Shrimali, Josef T Prchal, Prasenjit Guchhait.

**Writing – review & editing:** Garima Joshi, Kanchan Bhardwaj, Tsewang Chorol, Tashi Thinlas, Parvaiz A Koul, Prasenjit Guchhait.

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
