## [Decision Letter · Decision Letter 0]

PPATHOGENS-D-24-02260

Tibetan PHD2D4E;C127S variant protects from viral diseases in hypoxia, but predispose to infections in normoxia via HIFα:IFN axis

PLOS Pathogens

Dear Dr. Guchhait,

Thank you very much for submitting your manuscript ""Tibetan PHD2D4E; C127S variant protects from viral diseases in hypoxia, but predisposes to infections in normoxia via HIFα:IFN axis" (PPATHOGENS-D-24-02260) for review by PLOS Pathogens.

Your manuscript was fully evaluated at the editorial level and by independent peer reviewers. The reviewers appreciated the attention to an important problem but raised some substantial concerns about the manuscript as it currently stands. These issues must be addressed before we would be willing to consider a revised version of your study.

We cannot, of course, promise publication at that time. We therefore ask you to modify the manuscript according to the review recommendations before we can consider your manuscript for acceptance.

Your revisions should address the specific points made by each reviewer.

I am returning your manuscript with three reviews. The reviewers came to different conclusions about the paper, as you will see. After reading the reviews and looking at the manuscript, I recommend Major Revision, since a large number of experiments would be required to address all the comments appropriately. I am sorry I cannot be more positive at the moment; however we are looking forward to receiving your revision.

When you are ready to resubmit, please be prepared to provide the following: (1) A letter containing a detailed list of your responses to the review comments and a description of the changes you have made in the manuscript. (2) Two versions of the manuscript: one with either highlights or tracked changes denoting where the text has been changed; the other a clean version (uploaded as the manuscript file).

We hope to receive your revised manuscript within 60 days. If you anticipate any delay in its return, we ask that you let us know the expected resubmission date by replying to this email. Revised manuscripts received beyond 60 days may require evaluation and peer review similar to that applied to newly submitted manuscripts.

Sincerely,

Saumitra Das

Academic Editor

PLOS Pathogens

orcid.org/0000-0002-0640-3586

Ashley L. St. John

Section Editor

PLOS Pathogens

Kasturi Haldar

Editor-in-Chief

PLOS Pathogens

orcid.org/0000-0001-5065-158X

Grant McFadden

Editor-in-Chief

PLOS Pathogens

orcid.org/0000-0002-2556-3526

**Journal Requirements:**

At this stage, the following Authors/Authors require contributions: Riya Ghosh, Garima Joshi, Nishith M Shrimali, Kanchan Bhardwaj, Tsewang Chorol, Tashi Thinlas, Parvaiz A Koul, Josef T Prchal, and Prasenjit Guchhait. Please ensure that the full contributions of each author are acknowledged in the "Add/Edit/Remove Authors" section of our submission form.

Potential Copyright Issues:

- Figure 8;  Please confirm whether you drew the images / clip-art within the figure panels by hand. If you did not draw the images, please provide a link to the source of the images or icons and their license / terms of use; or written permission from the copyright holder to publish the images or icons under our CC BY 4.0 license. Alternatively, you may replace the images with open source alternatives. See these open source resources you may use to replace images / clip-art:

5) We note that your Data Availability Statement is currently as follows: "Data will be available from the corresponding authors via email". Please confirm at this time whether or not your submission contains all raw data required to replicate the results of your study. Authors must share the “minimal data set” for their submission. PLOS defines the minimal data set to consist of the data required to replicate all study findings reported in the article, as well as related metadata and methods (https://journals.plos.org/plosone/s/data-availability#loc-minimal-data-set-definition).

- The points extracted from images for analysis..

6) Please amend your detailed Financial Disclosure statement. This is published with the article. It must therefore be completed in full sentences and contain the exact wording you wish to be published. Please ensure that the funders and grant numbers match between the Financial Disclosure field and the Funding Information tab in your submission form. Note that the funders must be provided in the same order in both places as well.

**Reviewers' Comments:**

Reviewer's Responses to Questions

**Part I - Summary**

Reviewer #1: The authors characterise the impact of a mutant form of PHD2 with a high prevalence in Tibetans previously described to possess a higher affinity for O2 compared to the Wt form resulting in enhanced degradation of HIF-alpha subunits even in reduced oxygen conditions.

Given the above when examining HIF protein expression by immunoblot in normoxic conditions elevated levels of HIF-1 are observed (Figs 2K,3B,5D,J) in cells possessing the mutant form of PHD2 as well as in control wells. Did the authors perform viability assays to confirm the health of the cells in all experiments as HIF expression in otherwise normoxic conditions is symptomatic of poor cell culture health and/or over confluency. Further the increased expression of HIF in PHD2-mutant cells seems incongruent with its reported increased hydroxylation activity, as one would expect decreased expression of HIF not elevated levels at baseline. Could the authors validate whether levels of hydroxylated HIF are indeed augmented by the mutant form of PHD2?

Further I have concerns over the approach taken by the authors to overexpress the Tibetan variant of PHD2 (as described in Sfig2). They appear to have transiently knocked down the endogenous form of PHD2 by siRNA followed by overexpression of the variant form. Their siRNA knockdown show a partial reduction in endogenous PHD2 which raises concerns that their subsequent overexpression would resulting in a mixed population of both Wt and mutant PHD2 in the same cell population. The authors should either confirm the relative levels of either form or better yet use CRISPR to generate PHD2 KO cells as is widely reported and then over express either Wt or Mutant PHD2. This would provide a much ‘cleaner’ system to study the specific impact of this mutant form of PHD2. The data presented using the system is in my opinion confounded by their approach raising concerns over the validity of their findings with their exisiting overexpression models. Whilst the PBMC data presented in figure 1 is compelling these findings need to be validated in a more scientifically robust genetic model.

Finally the animal experiment in Sfigure 3, lacks key parameters such as pathology scores and/or immunohistochemistry for viral antigen in lung sections as well as infectious viral titres. The authors also present little data validating the inhibitor used resulted in the expected effect in terms of HIF target gene expression. Whist modulation of innate response genes are observed (Fig5 J-P) no functional validation is provided that the inhibitor is reducing HIF-target gene expression or inhibiting HIF activity. As presented, the phenotypes observed cannot be ascribed to inhibition of HIF activity without further evidence the inhibitor is working as expected.

Reviewer #2: In the manuscript entitled "Tibetan PHD2D4E;C127S variant protects from viral diseases in hypoxia, but predisposes to infections in normoxia via HIFα:IFN axis" by Ghosh et al., authors have investigated the role of the Tibetan-specific variant of prolyl-hydroxylase-2 (PHD2) D4E;C127S in immune response protection against dengue virus infection. They demonstrated that monocytes expressing PHD2-D4E;C127S promote IRF3-dependent production of IFNα/β under hypoxic conditions by suppressing HIF1α expression. Conversely, under normoxic conditions, these mutant cells fail to suppress HIF1α, leading to decreased synthesis of IRF3/IFNs and increased DENV2 infection. A similar pattern was also observed in the context of SARS-CoV-2 infection. The authors further confirmed the crosstalk between the PHD2D4E;C127S variant and the HIF1α-IFN axis using a HIF1α inhibitor and by expressing the variant in CD4/CD8-T cells. While the findings presented in the manuscript are intriguing, the poor explanation of results and concerns regarding data presentation and the timing of data analysis necessitate major revisions.

Reviewer #3: The manuscript by Ghosh et al describe experiments that show that the cells expressing the PHD2D4E;C127S variant of prolyl hydroxylase 2 are more susceptible to Dengue or SARS Cov2 infection in normoxia but show increased infection hypoxic conditions. The authors show that the PHD2D4E;C127S variant reduces Hif1 levels in hypoxic conditions and increases the interferon A and B pathway to regulate viral infection. While the results are interesting and of importance and suggest an adaptive advantage to the Tibetians, who carry this variant. The results seem to be a little unclear and incomplete to me as some of the key experiments and details of the quantitative methods are not optimal. It was previously known that the PHD2 variant has higher sensitivity to Oxygen and is capable of degrading Hif1 even when the pO2 levels are low. In the present manuscript the authors try to link the altered leves of Hif1 to the immune response against Virus in terms of interferon signalling.

**Part II – Major Issues: Key Experiments Required for Acceptance**

Reviewer #1: A large number of experiments would be required before the MS is suitable for publication - hence the recommendation to reject

Reviewer #2: o The authors should clarify the specific time point chosen for DENV2 infection and subsequent analyses. Given the dynamic nature of HIF1α expression under hypoxic conditions, it is crucial to assess HIF1α levels both before and after viral infection to accurately evaluate its role in the infection process.

o In Figures 1K and 1L, the methodology used to identify HIF1α-positive cells should be detailed. Including a representative image of HIF1α staining, similar to Figure 1B, would provide a clearer understanding of HIF1α expression dynamics during infection.

o Figures 1K, 1L, and 2K demonstrate increased HIF1α expression in PHD2D4E;C127S variant cells under normoxia, attributed to elevated ROS levels. To further elucidate the mechanism underlying this upregulation, the authors should directly compare ROS levels and HIF1α expression between uninfected and virus-infected monocytic cells. This would help determine the specific source of ROS and its causal role in HIF1α activation.

o In Figure 4B, treatment with the HIF1α inhibitor (CAY10585) under normoxia enhances p-IRF3 expression in both EGLN1WT and EGLN1c(12C>G; 380G>C) cells, while total IRF3 protein levels remain unchanged. However, Figure 4G shows an increased IRF3 gene expression in EGLN1WT cells after CAY10585 treatment, with a more pronounced effect in EGLN1c(12C>G; 380G>C) cells. The authors should provide an explanation for the apparent discrepancy between gene expression and protein levels.

o The gain-of-function EGLN1c(12C>G; 380G>C) variant fails to inhibit HIF1α expression under normoxic conditions with elevated ROS levels. It is important to investigate whether ROS affected variant-mediated HIF1α hydroxylation or its subsequent ubiquitin-mediated degradation.

o While the study primarily focuses on HIF1α-mediated inhibition of the IRF-IFN axis, it is possible that other pro-inflammatory cytokines independent of HIF1α may contribute to protection against viral infection in Tibetan individuals. Have authors explored any such direction?

Reviewer #3: 1. In Fig3 The authors show an elevated ROS levels in variant compared to wt in normoxia upon viral infection it would be interesting to also assess the ROS levels without viral infection. Do the authors have any suggestions on how ROS levels are regulated differentially in these two variants

3. In Fig5 the KD of hif1 seems to have a very modest effect on the viral RNA levels in hypoxic conditions while the interferon gene stimulation seems to be as much as seen in other experiments. In addition, it would have been better to also show the effect of Hif1KD in normoxic conditions too. In Fig6 the experiments are performed only in normoxic conditions It would be better to show a contrast between normoxic and hypoxic conditions in this case too.

3. Very interesting data in Fig7. As it shows that IfnG positive CD$ and CD8 increases in in response to peptide challenge. The significance of the differences observed in cells without peptide should also be indicated as it seems that even without challenge of the peptide, the levels of the IfnG cells alters between the Wt and the variant. It would be interesting to see if these differences are also observed in individuals not exposed to the virus or vaccinated, although it would be a challenge to find such samples.

**Part III – Minor Issues: Editorial and Data Presentation Modifications**

Reviewer #1: Not applicable

Reviewer #2: o In the result section “An inhibitors to HIF1α rescued”, please correct the sentence.

Reviewer #3: 1. In figures expressing the virus RNA levels that Y axis mentions that as compared to the mock (I am assuming infection). I am not sure that is the correct way to express as a better way would be to express them as a ration of DENV RNA to house keeping gene, as the mock infected cells should not be having any DENV RNA.

2. Can the authors explain why the DENV RNA levels as per the graph is increased by 10 fold (1 log 10 unit) between PHD2 wt and the variant in normoxic conditions (Fig1A and Fig2A), while the MFI of the viral protein stain show only a modest increase of 50-100% (Fig1C and 2C)

3. Was the U937 and HuH7 cells sequenced to verify the alleles that are expressed in these two cell lines that are being used if not they need to be verified

4. For Hif1 inhibition the inhibitor conc. Used is 40 microM while a lower concentration is (10 microM) is typically used for inhibiting Hif1a. any reasons for using such high concentrations. Further it is not clear whether the cells were in hypoxic or normoxic conditions in these experiments, Ideally it should be performed in both conditions

PLOS authors have the option to publish the peer review history of their article (what does this mean? ). If published, this will include your full peer review and any attached files.

**Do you want your identity to be public for this peer review?** For information about this choice, including consent withdrawal, please see our Privacy Policy .

Reviewer #1: No

Reviewer #2: No

Reviewer #3: No

**Figure resubmission:**
---

## [Decision Letter · Decision Letter 1]

PPATHOGENS-D-24-02260R1

Tibetan PHD2D4E;C127S variant protects from viral diseases in hypoxia, but predispose to infections in normoxia via HIFα:IFN axis

PLOS Pathogens

Dear Dr. Guchhait,

Thank you for submitting your manuscript to PLOS Pathogens. After careful consideration, we feel that it has merit but all the comments are not appropriately addressed.

One of the Reviewers has expressed some concerns which needs to be addressed before acceptance.

Therefore, we invite you to submit a revised version of the manuscript that addresses the points raised. 

Please submit your revised manuscript within 30 days.

We look forward to receiving your revised manuscript.

Kind regards,

Saumitra Das, Ph.D.

Academic Editor

PLOS Pathogens

Ashley St. John

Section Editor

PLOS Pathogens

Sumita Bhaduri-McIntosh

Editor-in-Chief

PLOS Pathogens

orcid.org/0000-0003-2946-9497

Michael Malim

Editor-in-Chief

PLOS Pathogens

orcid.org/0000-0002-7699-2064

**Journal Requirements:**

1)  Please remove your Response to the Journal Requirements from your Cover Letter. 2) We have noticed that you have uploaded Supporting Information files, but you have not included a complete list of legends. Please add a full list of legends for your Supporting Information files (RAW DATA_FINAL 2.zip) after the references list. 3) Please amend your detailed Financial Disclosure statement. This is published with the article. It must therefore be completed in full sentences and contain the exact wording you wish to be published.1) State what role the funders took in the study. If the funders had no role in your study, please state: "The funders had no role in study design, data collection and analysis, decision to publish, or preparation of the manuscript.".

**Reviewers' Comments:**

Reviewer's Responses to Questions

**Part I - Summary**

Reviewer #2: (No Response)

Reviewer #4: The researchers examine the influence of a PHD2 variant (D4E;C127), previously recognized as an adaptive mutation in Tibetans living at high altitudes with restricted oxygen levels, on the cellular susceptibility to infections by dengue virus (DENV) or SARS-CoV-2. This variant demonstrates a stronger affinity for oxygen than the original protein and sustains its enzymatic activity under various low-oxygen conditions, which results in the degradation of the HIF-alpha subunit. Whilst the premise is intriguing there is too much ambiguity in the written text and concerns with the presented data to warrant publication without major revisions

**Part II – Major Issues: Key Experiments Required for Acceptance**

Reviewer #2: (No Response)

Reviewer #4: The PHD2 mutant (D4E;C127) has been previously reported to retain a higher affinity for oxygen and thus retain its hydroxylation activity in lower oxygen concentrations compared to WT, thus reducing HIF-1 stabilisation. Whilst this is observed in some experiments (fig3C), in nomoxia expression of the mutant PHD appears to result in HIF-1 stabilisation, which is more pronounced with DENV infection. The approach taken to express this mutant PHD is convoluted, shRNA depletion followed by over expression. What is lacking in all experiments is confirmation of the level of expression of the Wt or Mutant PHD2. This should be included as currently it is not clear whether the changes in HIF-1 expression are a result of incomplete depletion of the endogenous PHD2 or overexpression? Further expression of this mutated PHD2 should increase hydroxylation of HIF not reduce it as seen in fig 3F – can the authors explain this departure with the literature?

The data presented with DENV is contradictory. HIF is stabilised by DENV as reported in previous work, hypoxic culture increases replication and overexpression of the mutated PHD2 with increased hydroxylation capacity diminishes HIF and thus replication (figure 1). However this does not explain the results observed in normoxia where overexpression of the mutant PHD2 increases DENV replication. These figures lack HIF-1 or PHD1 expression data making it extremely hard to evaluate the validity of these results. Again it is possible that the depletion/over-expression approach employed here is resulting in contradictory findings. Importantly this is not discussed or mentioned in the text. To validate their findings a model is required with full genetic depletion of PHD2 via CRIPSR should be employed or alternatively CRISPR editing of the endogenous PHD2 to introduce the mutation.

The HIF staining pattern in Figure 1M is irregular, HIF-1 when stabilised should localise to the nucleus not the cytoplasm/perinuclear region.

For the HIF silencing experiments presented in figure 5- the level of HIF silencing in these specific experiments should be shown, not just a reference to the exemplar in figure S2.

I’m not convinced the SARS-CoV-2 infection data in figure 6 is helpful as the use of the Huh7 cell line is not optimal for this virus. At minimum a respiratory cell line should be employed. Have the authors considered removing this figure and just presenting figure 7?

**Part III – Minor Issues: Editorial and Data Presentation Modifications**

Reviewer #2: (No Response)

Reviewer #4: For all figures the data points should be shown

PLOS authors have the option to publish the peer review history of their article (what does this mean? ). If published, this will include your full peer review and any attached files.

**Do you want your identity to be public for this peer review?** For information about this choice, including consent withdrawal, please see our Privacy Policy .

Reviewer #2: No

Reviewer #4: No

**Figure resubmission:**
---

## [Editor Report · Decision Letter 2]

Dear Dr. Guchhait,

We are pleased to inform you that your manuscript 'Tibetan PHD2D4E;C127S variant protects from viral diseases in hypoxia, but predispose to infections in normoxia via HIFα:IFN axis' has been provisionally accepted for publication in PLOS Pathogens.

Best regards,

Saumitra Das, Ph.D.

Academic Editor

PLOS Pathogens

Ashley St. John

Section Editor

PLOS Pathogens

Sumita Bhaduri-McIntosh

Editor-in-Chief

PLOS Pathogens

orcid.org/0000-0003-2946-9497

Michael Malim

Editor-in-Chief

PLOS Pathogens

orcid.org/0000-0002-7699-2064
---

## [Editor Report · Acceptance letter]

Dear Dr. Guchhait,

We are delighted to inform you that your manuscript, "Tibetan PHD2D4E;C127S variant protects from viral diseases in hypoxia, but predispose to infections in normoxia via HIFα:IFN axis," has been formally accepted for publication in PLOS Pathogens.

Best regards,

Sumita Bhaduri-McIntosh

Editor-in-Chief

PLOS Pathogens

orcid.org/0000-0003-2946-9497

Michael Malim

Editor-in-Chief

PLOS Pathogens

orcid.org/0000-0002-7699-2064